# Study of Physico-Chemical Interactions during the Production of Silver Citrate Nanocomposites with Hemp Fiber

**DOI:** 10.3390/nano11102560

**Published:** 2021-09-29

**Authors:** Alexandru Cocean, Iuliana Cocean, Georgiana Cocean, Cristina Postolachi, Daniela Angelica Pricop, Bogdanel Silvestru Munteanu, Nicanor Cimpoesu, Silviu Gurlui

**Affiliations:** 1Atmosphere Optics, Spectroscopy and Laser Laboratory (LOASL), Faculty of Physics, Alexandru Ioan Cuza University of Iasi, 11 Carol I Bld, 700506 Iasi, Romania; alexcocean@yahoo.com (A.C.); iulianacocean@hotmail.com (I.C.); cocean.georgiana@yahoo.com (G.C.); tina.postolaki@gmail.com (C.P.); nicanornick@yahoo.com (N.C.); 2Rehabilitation Hospital Borsa, 1 Floare de Colt Street, 435200 Borsa, Romania; 3Faculty of Physics, Alexandru Ioan Cuza University of Iasi, 11 Carol I Bld, 700506 Iasi, Romania; daniela.a.pricop@gmail.com (D.A.P.); msbogdanel18@yahoo.com (B.S.M.); 4Faculty of Material Science and Engineering, Gheorghe Asachi Technical University of Iasi, 59A Mangeron Bld, 700050 Iasi, Romania

**Keywords:** nanocomposites, pulsed laser deposition

## Abstract

In the study presented in this paper, the results obtained by producing nanocomposites consisting of a silver citrate thin layer deposited on hemp fiber surfaces are analyzed. Using the pulsed laser deposition (PLD) method applied to a silver target with impurities of nickel and iron, the formation of the silver citrate film is performed in various ways and the results are discussed based on Fourier Transform Infrared (FTIR) and Scanning Electron Microscopy coupled with Energy Dispersive X-ray (SEM-EDX) spectroscopy analyses. A mechanism of the physico-chemical processes that take place based on the FTIR vibrational modes and the elemental composition established by the SEM-EDS analysis is proposed. Inhibition of the fermentation process of Saccharomyces cerevisae is demonstrated for the nanocomposite material of the silver citrate thin layer, obtained by means of the PLD method, on hemp fabric. The usefulness of composite materials of this type can extend from sensors and optoelectronics to the medical fields of analysis and treatment.

## 1. Introduction

Composite materials, based on texturized textile fibers—as a reinforcing phase, on which silver films are deposited as a continuous phase (matrix)—are made for use in medical devices and also for other applications where an ionic state of silver (Ag^+^) is required.

The aim of the experimental procedures presented in this work is to study if, during the pulsed laser deposition (PLD) process, the interaction of citric acid with the ablated silver, in its final plasma stage before hitting the support, and under vacuum chamber conditions, could lead to a chemical reaction that results in the formation of citrate. It is known that silver does not react with citric acid unless one of the reactants is in an ionic state (inorganic compounds of silver, such as AgNO_3_, would enter into a reaction with citric acid, or sodium citrate would enter into a reaction with silver). Moreover, silver does not react with cellulose in its metallic state. In order to react with cellulose, “silver seeds”, i.e., silver nitrate (AgNO_3_), and reagents are required [1]. Other methods utilized for producing silver nanoparticle synthesis in an ionic state, for medical purposes, are based on so-called “green synthesis”, where different biological organisms or extracts of those participate in a photochemical or chemical process [2,3,4,5,6,7,8]. Plasma reduction is another method used to synthesize Ag nanoparticles and Pt nanoparticles for further applications in dye-sensitized solar cells [9,10].

The impetus of this research, involving the production of compounds in which silver is in an ionic state, derives from its proven benefits regarding its antibacterial and antifungal properties. Researchers are also concerned with possible applications of gold nanoparticles in cancer treatment, where gold nanoparticles in an ionic state are supposedly the active form against cancer cells. In this respect, synthesis of gold nanoparticles has been achieved in the form of gold citrate but also as gold ionic bonded to polymers such as polyethylene imine [11]. Solar-driven water evaporation could be another application of such developed materials [12,13].

In the general context of interest for the study of noble metal nanoparticles, the method presented in this paper proposes both a new method of producing silver citrate embedded in a composite of the hemp-reinforcing phase, and a model of the physico-chemical mechanism of the process that takes place during, and under the conditions of, the pulsed laser deposition.

## 2. Materials and Methods

Pulsed laser deposition (PLD) was performed on the installation in the Atmosphere Optics, Spectroscopy and Lasers Laboratory [14] using the YG 981E/IR-10 laser system, with the parameters τ = 10 ns pulse width, λ = 532 nm wavelength, α = 45° incident angle and ν = 10 Hz pulse repetition time and 3·10^−2^ Torr pressure, in the deposition chamber (Figure 1). The target subject for ablation (Figure 2) was made of silver with iron and nickel impurities. It was produced from jewelry scraps by means of mechano-thermic processes and chemical cleaning with 99.9% tetra borate (Na_2_B_4_O_7_·10H_2_O) and 99% sodium borohydride (NaBH_4_) to convert silver from the ionic to the atomic state [15,16] followed by treatment with baking soda as a catalyst (NaHCO_3_ 99.5%) in the presence of aluminum foil in order to remove sulfur from the silver sulfide (Ag_2_S).

The experimental studies refer to the physico-chemical interaction of plasma plume with the substrate containing citric acid. For that purpose, a total of three samples were produced, as follows:The film deposition on hemp fabric is noted as sample *D* (*Ag Film/Hemp*).The next film was deposited on a hemp fabric impregnated with supersaturated aqueous citric acid solution to form composite materials for further applications; this is referred to as sample *E* (*Ag Film/CA/Hemp*).Finally, silver film deposition was performed on a layer of citric acid applied as a supersaturated aqueous citric acid solution on a glass slab; this is denoted as sample *F* (*Ag Film/CA/Glass*).

The fabrics used as supports are noted as follows: HTND—the hemp twill fabric (natural); HTNS—the hemp twill fabric (natural) impregnated with citric acid solution.

Pulsed laser deposition was performed with laser energy of 150 mJ on a spot with a 168 µm average standard deviation, with a distance target support of 2 cm. The deposition time was 30 min for 18 × 10^4^ pulses, the number of pulses necessary for obtaining a consistent silver layer [17].

A test to demonstrate the functional property of the new nanocomposite material, Ag-CA-HMP, related to inactivation of the microorganisms, was conducted using dry baker’s yeast (Saccharomyces cerevisiae). Samples of hemp fabric, hemp fabric coated with a silver layer and hemp fabric coated with a silver citrate layer, obtained using the PLD method as described in the study presented herein, were placed on three glass slabs of 22 mm × 22 mm size, while one glass slab was used as a reference (R) or for the blind test. Each of the four glass slabs was placed in a different Petri dish. Quantities of 20 mg of yeast mixed with 10 mg of sugar were placed on the center of each sample and 0.5 mL of distilled water, at 45 °C–50 °C, was added on the mixture of yeast and sugar using a pipette. The samples obtained this way were named as follows:*1-Y-active* (yeast mixture on the glass slab or blind sample).*2-Y-HMP* (the yeast mixture on hemp fabric sample).*3-Y-Ag-HMP* (the yeast mixture on the silver layer deposited on the hemp fabric).*4-Y-Ag-CA-HMP* (the yeast mixture on the silver citrate layer deposited on the hemp fabric).

The reference or blind sample was noted as *R-Y-DRY*.

The surfaces and the foaming bubbles of CO_2_ were measured in pixels using the Toup View software.

## 3. Results and Discussions

### 3.1. Target Initial Chemical Composition

After the preparatory steps of the target, EDX analysis showed a composition, in atomic percentages, of 81.84% Silver, 17.40% Nickel, and 0.76% iron in some areas; 88.26% Silver, 10.32% Nickel, 0.44% iron in other areas; and even 100% Silver in some areas. The composition demonstrated a non-homogenous distribution of the impurities through the silver matrix of the target, which is consistent with the usual conditions of manufacturing where materials of pure elements are not used due to their high cost and difficulties involved in purifying them, as well as the special storage conditions needed in order to avoid contamination/impurification.

### 3.2. Physico-Chemical Processes of Ablation and Redisposition on the Target

Analyzing the ablated area on the target after it was used in the PLD process, an important increase in iron and nickel was noticed compared with the results obtained before ablation. Thus, the elemental compositions of two areas, of 0.053 mm^2^ each, that were analyzed at the center of the target after ablation (Figure 3b,c) were found as being of 64.56% Silver, 27.01% Nickel, 8.43% Iron and 77.20% Silver, and of 17.56% Nickel, 5.20% Iron, respectively. The detected elements were evidenced by the EDS spectrum presented in Figure 4. The increase in Ni and Fe atoms on the ablated area of the target was the result of different processes and phenomena during ablation such as re-deposition (Figure 3a–c) of the lighter elements (atomic mass, A, and atomic number, Z, of each component being N2858.69i where A_Ni_ = 58.69, F2655.845e where A_Fe_ = 26 and A47108g where A_Ag_ = 108) that may have been due to their elastic collision with heavier atoms, ions and clusters, but also due to other perturbing phenomena including, but not limited to, the influence of electromagnetic fields on the charged particles (ions). Preferential Silver ablation was indicated with increase in Ni and Fe in the elemental composition of the ablated area, which is important for the quality of the deposited thin film.

In the SEM images of Figure 3a–c, the droplets of the re-deposited material can be noticed on the ablated area.

### 3.3. Physico-Chemical Interaction of Silver Plume with the Citric Acid Substrate

The SEM images (500 × magnifying) of the three Silver film deposition samples are presented in Figure 5. The silver layer was not only deposited on the fabric surface. It penetrated through the interstices between the twisted fibers that formed the yarns from the fabric texture. In this way, the deposition produced a composite structure with new properties. The ionic state of the silver was the intended goal due to its antibacterial and antifungal properties and, for that reason, the study of the interaction of silver plasma plume with the citric acid from the support was important.

The images in Figure 5 evidence the droplets of silver resulted during the laser deposition. EDS analysis showed an elemental composition on the thin layers as presented in Table 1 and in the spectra of Figure 6a–c. The analyzed areas were of 0.185 mm^2^ on the surfaces presented in Figure 5b–d) for data in Table 1.

The PLD technique is based on laser ablation when the solid material of the target is transformed into liquid, gas and plasma. Plasma, also known as ionized gas [18,19,20,21,22,23,24,25,26], is a state of matter where ions, electrons, atoms and clusters exist at the same time. The predominant content in the Silver of the deposited thin film is noted in Table 1 This is in good accordance with the results of elemental composition of the ablated area on the target. It implies that the silver ions should arrive on the support surface and react with the citric acid. Because some of the silver was already derived from the ablation that occurred in liquid state, as per the temperature plots obtained from the COMSOL simulation of silver ablation and depositions shown in the work of A. Cocean et al., 2017 [27], the reaction rate was not expected to be high. Nevertheless, other phenomena can improve the active state of the film and increase its predisposition to subsequent reactions when in contact with biological agents such as bacteria and fungi. Regardless, this was not the experimental subject for this study, but it was where the idea started. Therefore, in order to determine whether the reaction took place, the powder of the deposited layers was collected by scraping it off the surface of the samples, and the FTIR results are presented in Figure 7. In order to avoid the possibility that silver ions could occur from oxidized areas or from Ag_2_S that may form, in time, on the target surface when in contact with the atmosphere, borax was incorporated on the target (as presented in the Section 2). It followed that silver ions from oxides and/or sulfides had been converted into atoms before the PLD process, but also during PLD if any remaining oxygen traces were present in the deposition chamber. This provides further evidence that the only source of silver ions that will react with the citric acid is the plasma obtained via ablation during which ions, atoms and clusters co-exist. For a better evaluation of the FTIR spectra of the samples that were experimentally obtained, they were also compared with the spectra of the hemp fabrics used as supports for silver deposition, namely HTND (hemp fabric) and HTNS (hemp fabric impregnated with the supersaturated citric acid aqueous solution). The FTIR spectra (Figure 7) show the changes that citric acid undergoes as a result of silver deposition. The transformation of carboxyl groups into carboxylate ions can be observed in the FTIR spectra, and the formation of silver citrate is indicated.

In this regard, for both types of hydroxyl groups assigned to carboxyl from citric acid (free: 3500 cm^−1^; H-bounded: 3283 cm^−1^ [28,29]), there is evidence that they were transformed into ionized COO^−^ groups, which were involved in ionic bonds with Ag^+^, as indicated by the carbonyl bands from 1638 cm^−1^ to 1760 cm^−1^ [28,29].

The bands from about 3500 cm^−1^ were the alcoholic groups [28,29] of the citric acid spectrum (CA), but they were also found in the sample of PLD film spectra for citrate when the peak became broader due to hydrogen bonding. This caused the silver ions from the plasma that was produced via ablation, that still existed when arriving on the substrate surface, to enter into a reaction with the citric acid, and thus, may have formed trisilver citrate as well as also mono- and disilver citrate (Figure 8). However, the physico-chemical process was more complex than only a citric reaction with silver plasma ions. The presence of intermolecular H-bonds between the hydroxyl groups of the citrate is indicated in the broad band at 3500 cm^−1^ in the sample E spectrum. Furthermore, it has been established that the partially ionized oxygen from the carbonyl groups will enter into hydrogen bonds with hydroxyl groups from other molecules, but also intramolecular bonds, as the peak from 1638 cm^−1^ could indicate [28,29]. H-bonding could also have occurred between the citrate and cellulose (Figure 9b). The spectrum of sample E (hemp impregnated with citric acid before deposition) evidences the formation of citrate. As for the silver that was deposited directly on the citric acid (sample F), the spectrum indicates that some of the citric acid had reacted with silver plasma ions, while some still existed as citric acid, or alternatively, only part of the carboxylic groups had been transformed into carboxylates (Figure 9c).

Due to the environmental conditions in the vacuum chamber, it was possible for a further ionized structure of the silver citrate to form, which may have interacted through ionic and H-bonding and formed aggregates and complex structures, the most evident being for the citrate formed on the glass slab. The very broad band of the F (Ag Film/CA/Glass) spectra between 3560 and 2598 cm^−1^ indicates that both the carboxyl and carboxylate groups coexisted [28,29], meaning that part of the citric acid was transformed into citrate and part remained as citric acid. The broad band also indicates H-bonds, intermolecular ionic bonds and other intermolecular interactions, such as Van Der Waals interactions. Such a model may be depicted as in Figure 9a–c. Van Der Waals interactions between ionized and/or partially ionized atoms would have taken place in both films (the E-film on the hemp treated with citric acid and the F-film that was deposited on the citric acid). The adsorption of silver atoms on the carbonyl groups was also part of the complex interactions in the thin film system (Figure 9b,c).

The reactions, namely the intermolecular and intramolecular interactions, that may occur upon the impact of silver plasma with the substrate surface containing citric acid are schematically presented in Figure 8 and Figure 9a–c. Further studies of this process of interaction between the silver plasma ions and the citric acid molecules during pulsed laser deposition, and its mechanism, may lead to a technique that can produce silver layers that are active against bacteria as well as being useful for other applications where an ionic state of silver and/or other metals is required. As noticed when analyzing the FTIR spectrum for silver deposited, by means of the PLD technique, on the citric acid, not all of the citric acid was transformed into citrate; this was because the quantity of the citric acid was excessive compared to the quantity of silver ions formed in the plasma. Based on this observation, a method to measure the quantity of the ions of metals that arrive on the substrate, and further, a PLD method of “titration”, could be developed to analyze different intermediary compounds formed during the travelling of plasma on the path from the target to the substrate.

Regarding the sizes and shapes of the nanoparticles, UV-Vis spectral analysis was performed on sample F (Ag Film/CA/Glass). For the analysis, material was scraped from the upper part of the layer and was deposited on a glass slide (Ag Film/CA/Glass), with care taken not to scrape the glass directly. The material was dissolved in 2 mL of distilled and deionized water. The distilled and deionized water was used for comparison. Because the amount of material taken from the thin layer for UV-Vis analysis was very small and the dilution was high (imposed by the 2 mL cuvette volume), the signal obtained for absorption was of low intensity. However, peaks that, in the literature, were assigned to various sizes of silver nanoparticles, as well as silver citrate, can be observed. Thus, the UV-Vis spectrum (Figure 10) shows a succession of 284 nm, 353 nm, 412 nm, 443 nm peaks and a wide band between 487 nm and 543 nm. The peak at 284 nm could be related to the formation of (Ag^+^)_3_/citrate complexes [30], probably in the form of silver clusters and/or ultra-small silver nanoparticles [31]. Furthermore, in the same spectrum, a narrow band at 353 nm and a wide band between 487 and 543 nm appeared, possibly caused by the fusion of heated nanoparticles as a result of laser exposure. The spectral footprint of the fused nanoparticles is suggested by the wide band (between 487 and 543 nm) of the longitudinal oscillation mode of the surface plasmon and the narrow band at 353 nm of the transverse oscillation mode [32,33]. The fused nanoparticle formation could also be assigned to the broader bands, at 3500 cm^−1^, of the FTIR spectra of samples E (AgFilm/CA/Hemp) and F(AgFilm/CA/Glass), but without neglecting other forms of aggregation, such as those described and presented in Figure 9.

Two other peaks at 412 nm and 443 nm can be attributed to the formation of triangular nanoparticles, with dimensions of around 40 nm [34], and nanoparticles with uneven shapes between 60 and 70 nm [35].

Citrate molecules act as both a capture ligand for silver particles and as a photoreducing agent for silver ions [36,37]. It is known that under the action of light radiation, citrate photoresists silver ions at the surface of previously formed silver seeds. The growth of nanoparticles with different morphologies is due to the rates of citrate reduction on certain faces of silver crystallites [34]. Silver–citrate complexes formed as a result of pulsed laser deposition indicate a possible application of the developed material as an antimicrobial layer [38].

### 3.4. Yeast Foaming Test for the New Ag-CA-HMP Material Synthesized by PLD Method

During the foaming test described in the Section 2, an intense foaming activity was observed on sample 1-Y from the first moment; the yeast on samples 2-Y-HMP and 3-Y-Ag-HMP was less active, while there was no foaming activity on sample 4-Y-Ag-CA-HMP (Figure 11).

Based on the measurements of the samples and the dimensions of the glass slabs made using the Toup View software, in addition to the diameters of the CO_2_ bubbles, the foaming activity (FA%) and foaming stability (FS%) were calculated for the initial moment of the foaming test, and at 3 min from the point at which the foaming test had started. The foaming activity and foaming stability were calculated as follows:

Glass slab area:A=L·l

The areas of samples 2, 3 and 4 (the samples on the fabric have one quarter of the area of the circles seen in Figure 11):$$A=\frac{\pi \cdot R^{2}}{4}=\frac{\pi \cdot D^{2}}{16}$$

Areas of the bubbles:$$a=\pi \cdot r^{2}=\frac{\pi \cdot d^{2}}{4}$$

The total foam area on each sample as the sum of the areas of the bubbles:$$F=\sum_{i=0}^{n}a_{i}$$

The foaming activity, as a percentage of the foam area from the sample area and from the slab area, respectively:$$FA\%=\frac{F}{A}\cdot 100$$

The foaming stability, as a percentage of the foaming activity after 3 min from the point at which the foaming test was started, derived from the initial foaming activity of each sample:$$FS\%=\frac{FA_{initial}\left(\%\right)}{FA_{after\;3\;minutes}\left(\%\right)}\cdot 100$$

The results are presented in Table 2 and in Figure 12.

The changes that occurred within 3 min of the initiation of foaming activity among the studied samples, using the surface of each sample and that of the glass slab on which the sample was placed as references, are presented in Figure 12a,b. The variations in foaming stability among the studied samples, using the area of each sample and that of the glass slab, are shown in Figure 13a,b.

The samples that were produced via the foaming test were dried at 25 °C for 24 h, and each sample was kept in a Petri dish. After drying, yeast material was collected from the glass slab of each sample. Each sample of yeast material was mixed with KBr in a mortar and, after that, was pressed into a ring and FTIR analysis was performed. The same process was undertaken with the sample of dry yeast that was used in the foaming test. The resulting spectra of the four samples and of the reference sample (dry yeast) are presented in Figure 14a,b.

The peaks that were specific to the FTIR spectrum of the dry yeast sample (Figure 14, Y-DRY) were also found in the spectra of the Y-active, Y-HMP and Y-Ag-HMP samples, with very small variation in terms of their intensity. The 3417 cm^−1^ band may be assigned to N-H stretching in the same range as the O-H-free and H-bonded samples. The bands assigned to proteins (the band at 1631 cm^−1^ assigned to CNO stretching and N-H bending in amides I, and the band at 1545 cm^−1^ assigned to C-N stretching and N-H bending in amides II) [39,40], as well as those assigned to sugars (1057 cm^−1^ and 908 cm^−1^; these bands were specific to the cyclic ethers in carbohydrates) [39,40] or nucleic acids, denoted by the band at 1240 cm^−1^ as per A. Gallichet et al., 2001 [41], do not appear to have been essentially modified in the FTIR spectra of the mentioned samples.

Unlike the other three samples (Y-active, Y-HMP and Y-Ag-HMP), the test conducted the on Y-Ag-CA-HMP sample shows essential changes in the FTIR spectrum (Figure 14, Y-Ag-CA-HMP) compared to the spectrum of the dry yeast, and also in comparison to all other samples (Y-active, Y-HMP and Y-Ag-CA-HMP). In essence, the spectrum of the Y-Ag-CA-HMP sample shows the characteristics of sucrose (table sugar) even if some peaks appear to have overlapped those of yeast, which is also known to contain sugars.

Of note is the strong and sharp vibrational band from 3558 cm^−1^, which is specific to the spectrum of sucrose and is attributed to the free OH groups [41,42].

In the finger-print area of the Y-Ag-CA-HMP sample spectrum, the bands from 1403 cm^−1^, 1343 cm^−1^, 1275 cm^−1^, 1240 cm^−1^, 1207 cm^−1^, 1128 cm^−1^, 1066 cm^−1^, 1052 cm^−1^, 989 cm^−1^, 940 cm^−1^, 908 cm^−1^, 864 cm^−1^, 848 cm^−1^, 728 cm^−1^, 728 cm^−1^, 680 cm^−1^ and 663–565 cm^−1^ are highlighted, and these are assigned to ethers (C–O–C stretch asymmetric; arC–O–alC) and to related components [14,28,29]; these bands can also be found in online databases on sucrose (1427 cm^−1^, 1343 cm^−1^, 1279 cm^−1^, 1240 cm^−1^, 1208 cm^−1^, 1126 cm^−1^, 1065 cm^−1^, 1049 cm^−1^, 988 cm^−1^, 942 cm^−1^, 908 cm^−1^, 866 cm^−1^, 849 cm^−1^, 732 cm^−1^, 687 cm^−1^ and 640–521 cm^−1^) [39,40].

Based on the FTIR spectra shown in Figure 14, the interpretation of the foaming test results presented in Figure 12 and Figure 13, as well as in Table 2, is that the fermentative effect of the yeast (Saccharomyces cerevisiae) on sugar was inactivated by the silver citrate layer that was obtained on the hemp fabric using the PLD technique and the method reported in this paper.

Thus, in the case of the three samples (Y-active, Y-HMP and Y-Ag-HMP), the fermentation of sugar is highlighted both by the CO_2_ release process that can be observed in the foaming effect that occurred with the generation of the measured gas bubbles (Figure 12 and Figure 13 and Table 2) and by the FTIR spectra, which was similar to that of the initial yeast spectrum. In the case of the Y-Ag-CA-HMP sample (the silver citrate layer deposited on the hemp fabric that we fabricated according to the new method described herein), no CO_2_ bubbles were released during the foaming test, and in the immediate vicinity, there was only an insignificant amount of initial foaming activity (FA%) of 0.34%, with 0% foaming stability (FS%), at 3 min from initiation. This is in accordance with the FTIR spectrum of sample Y-Ag-CA-HMP, which also shows that the fermentation of sugar did not take place.

## 4. Conclusions

The ionic state of silver is important for the production of composites with antibacterial and antifungal properties for various applications, including medical, but its importance is not limited to them. This study has shown that the PLD technique is suitable for the production of silver citrate as a result of the interaction of the ablation plume with a citric acid substrate or a substrate with citric acid contents. There are also indications resulting from the experiment presented herein that a method of “titration” may be developed in order to determine the quantity of metals ionic state in the plasma plume at different distances from the target, based on specific reactions. The method could be extended to other materials, with the aim of producing silver-based composites for later applications in industry and medicine, as well as for environmental investigations and air and water filtration. The tests and investigations show that the silver citrate that was obtained on the hemp fabric using our new method inactivated the yeast (Saccharomyces cerevisiae), confirming the effect of silver ions on the microorganisms, which, at the very least, inactivated their metabolism. This proves that a new functional material was obtained. Such nanocomposites of the silver citrate layer on the hemp fabric can be useful in processes where fermentation or excessive foaming induced by yeasts needs to be controlled by inhibition processes. Further investigations are needed and will be conducted to study the antimicrobial effect of the new nanocomposite material.

In order to determine the oxidation state of silver during deposition and on the deposited layer, in our further studies, we will consider the development of a method of analysis to be utilized in the deposition chamber. This method is necessary because the analysis would take place without the risk of contamination of the sample as a result of contact with air.

## Figures and Tables

**Figure 1 nanomaterials-11-02560-f001:**
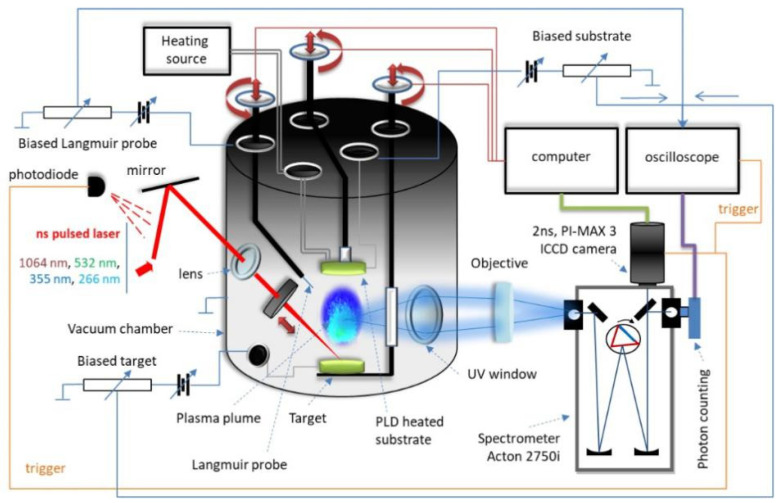
Experimental installation for PLD and LIBS in the Atmosphere Optics, Spectroscopy and Lasers Laboratory (http://spectroscopy.phys.uaic.ro. Accessed on 4 September 2021).

**Figure 2 nanomaterials-11-02560-f002:**
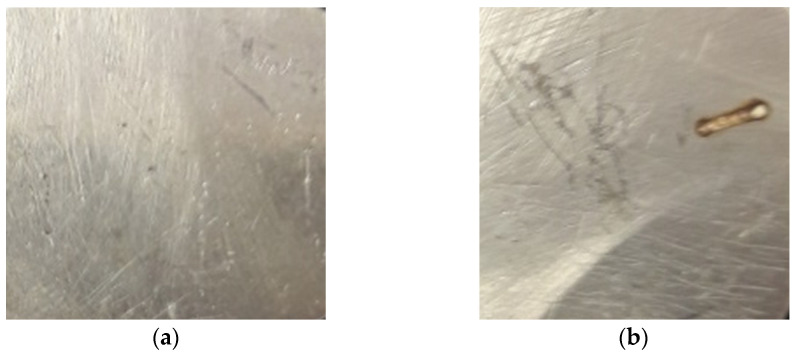
Silver target: (**a**) before irradiation; (**b**) after laser ablation during PLD processes.

**Figure 3 nanomaterials-11-02560-f003:**
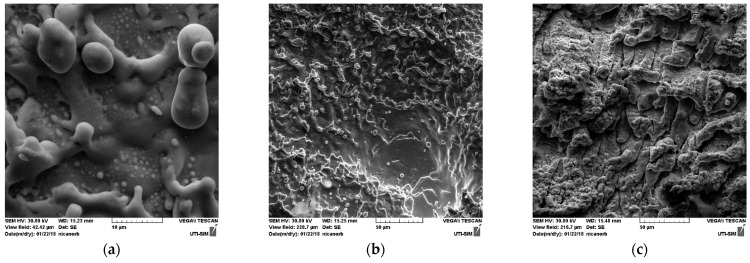
SEM images on ablated zone of the target with re-deposition structures: 5 kx magnitude (**a**); 1 kx magnitude (**b**,**c**).

**Figure 4 nanomaterials-11-02560-f004:**
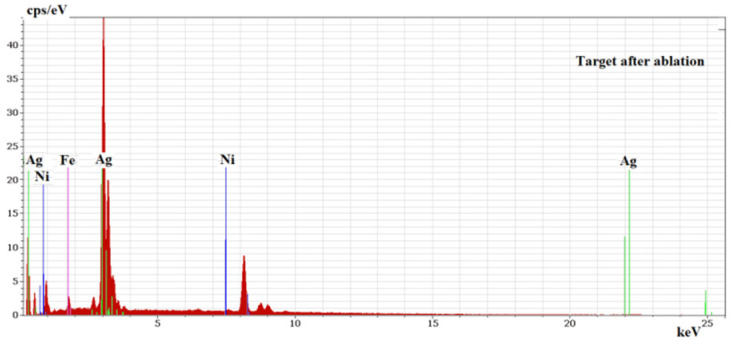
EDX spectrum of the target after ablation.

**Figure 5 nanomaterials-11-02560-f005:**
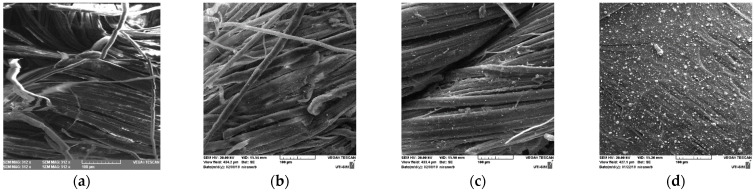
SEM images 500 x of the samples (**a**) Hemp fabric used as support for pulsed laser deposition; (**b**) Sample D (Ag Film/Hemp); (**c**) Sample E (Ag Film/CA/Hemp); (**d**) Sample F (Ag Film/CA/Glass).

**Figure 6 nanomaterials-11-02560-f006:**
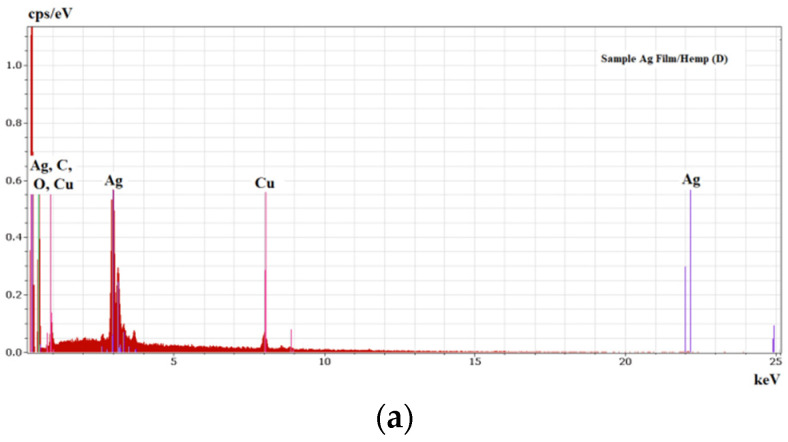
EDS spectra of the thin layers obtained by PLD: sample Ag Film/Hemp (D) (**a**); sample Ag Film/CA/Hemp (E) (**b**); sample Ag Film/CA/Glass (F) (**c**).

**Figure 7 nanomaterials-11-02560-f007:**
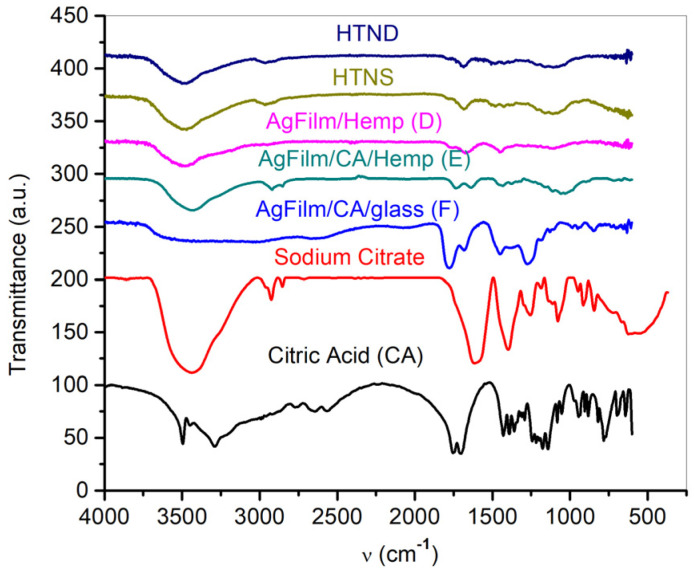
Comparison of the FTIR spectra of samples D (AgFilm/Hemp), E (AgFilm/CA/Hemp) and F (AgFilm/CA/glass), and the sodium citrate and citric acid (CA).

**Figure 8 nanomaterials-11-02560-f008:**
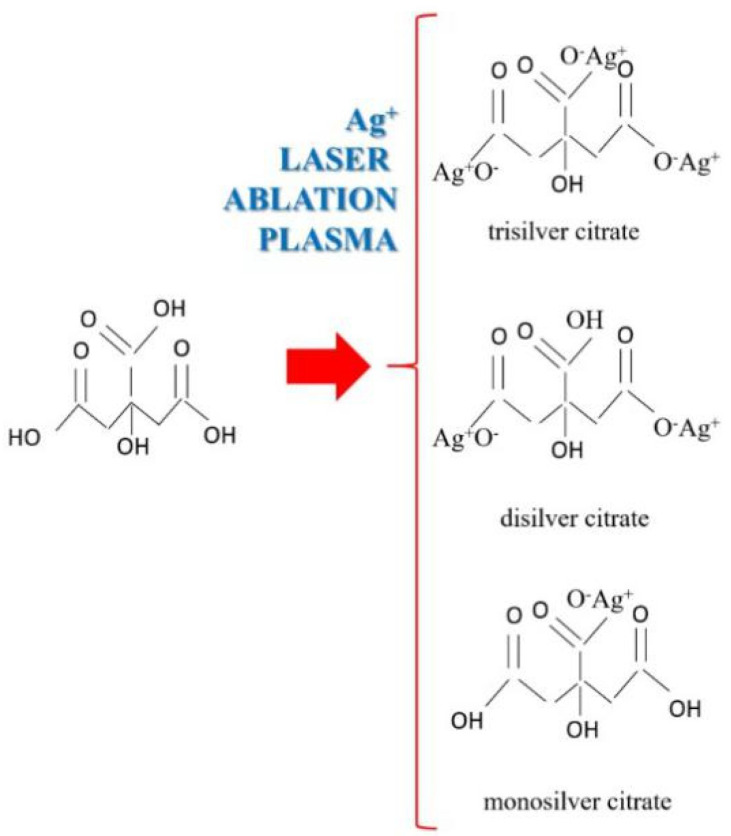
Chemical reaction of citric acid with silver ions from the laser ablation plasma plume.

**Figure 9 nanomaterials-11-02560-f009:**
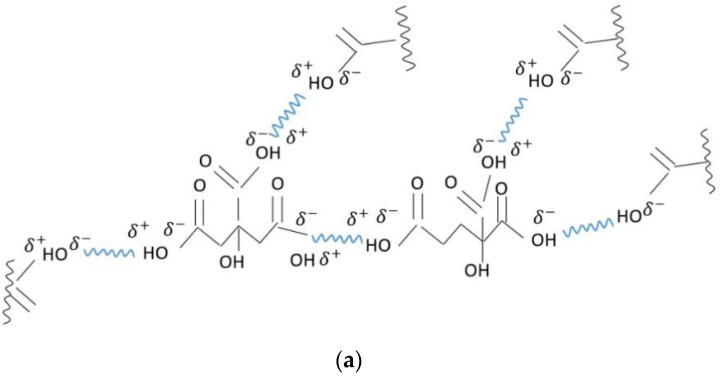
The reactions, namely the intermolecular and intramolecular interactions, that may occur upon the impact of silver plasma with the substrate surface containing citric acid: (**a**) hydrogen bonding in citric acid; (**b**) sequence of intermolecular H-bonding, inter- and intramolecular Van Der Waals interactions and Silver atoms’ adsorption on citrate and cellulose; (**c**) sequence of intermolecular H-bonding, inter- and intramolecular Van Der Waals interactions and Silver atoms’ adsorption on citrate.

**Figure 10 nanomaterials-11-02560-f010:**
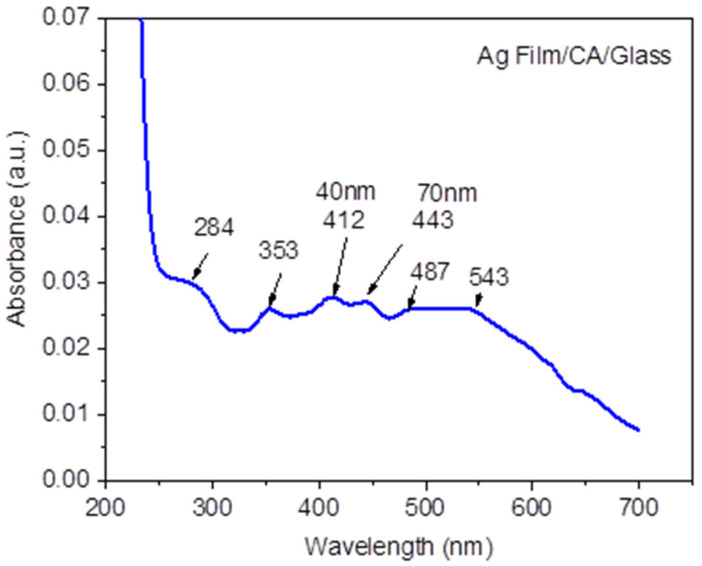
UV-Vis spectrum of sample F (Ag Film/CA/Glass).

**Figure 11 nanomaterials-11-02560-f011:**
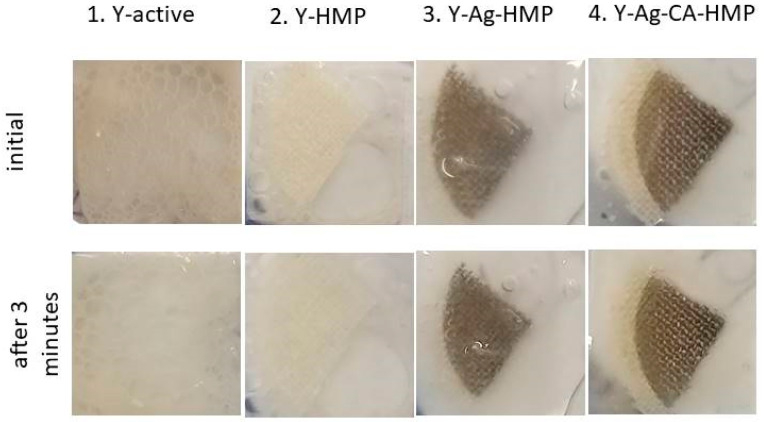
Images of foaming activity initial and after 3 min from starting the foaming test.

**Figure 12 nanomaterials-11-02560-f012:**
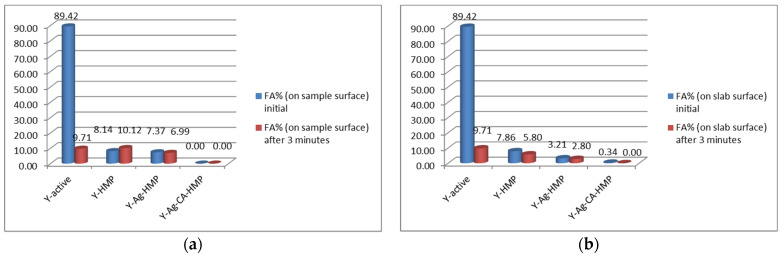
Comparison of yeast foaming activity (FA%) on the samples’ surfaces (**a**) and on slab surface (**b**).

**Figure 13 nanomaterials-11-02560-f013:**
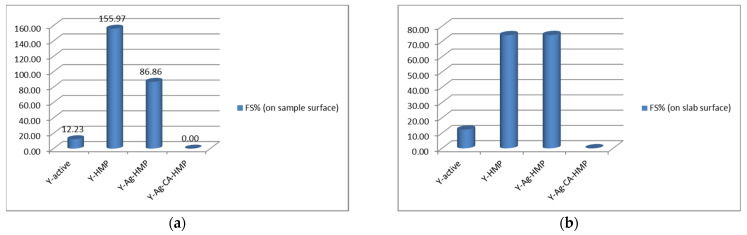
Comparison of yeast foaming stability (FS%) on samples’ surfaces (**a**) and on slab surface (**b**).

**Figure 14 nanomaterials-11-02560-f014:**
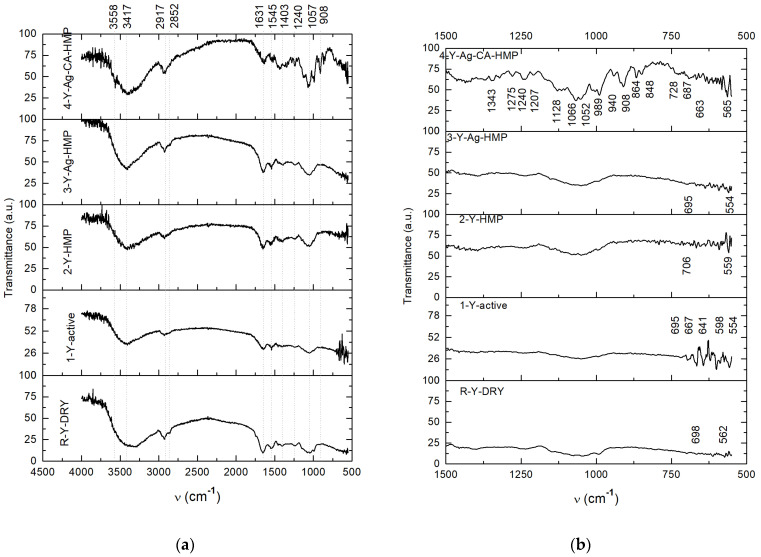
FTIR spectra of investigated behavior of yeast to different media provided by the analyzed materials (**a**) and detailed FTIR spectra in the finger-print region of 1500 cm^−1^–500 cm^−1^ (**b**).

**Table 1 nanomaterials-11-02560-t001:** Elemental composition of the samples: Ag Film/Hemp (D); Ag Film/CA/Hemp (E); Ag Film/CA/Glass (F).

Element	Norm. wt.%			Norm. at.%		
	Sample (D) Ag Film/ Hemp	Sample (E) Ag Film/ CA/Hemp	Sample (F) Ag Film/ CA/Glass	Sample (D) Ag Film/ Hemp	Sample (E) Ag Film/ CA/Hemp	Sample (F) Ag Film/ CA/Glass
Oxygen	76.46	76.98	52.47	82.47	82.90	86.98
Silver	10.72	10.41	41.06	1.71	1.66	10.09
Carbon	10.58	10.33	-	15.20	14.82	-
Copper	2.23	2.28	-	0.60	0.62	-
Nickel	-	-	6.46	-	-	2.92
	100	100	100	100	100	100

In the elemental composition of the samples D and E, copper and carbon belong to the hemp fabric.

**Table 2 nanomaterials-11-02560-t002:** Foaming activity (FA%) and foaming stability (FS%).

	Sample	FA% (on Sample Surface)		FA% (on Slab Surface)		FS% (on Sample Surface)	FS% (on Slab Surface)
		Initial	after 3 min	Initial	after 3 min		
1	Y-active	89.42	9.71	89.42	9.71	12.23	12.23
2	Y-HMP	8.14	10.12	7.86	5.80	155.97	74.09
3	Y-Ag-HMP	7.37	6.99	3.21	2.80	86.86	74.18
4	Y-Ag-CA-HMP	0.00	0.00	0.34	0.00	-	0.00

## Data Availability

Not applicable.

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
