# Peer review of "Study of Physico-Chemical Interactions during the Production of Silver Citrate Nanocomposites with Hemp Fiber"

_nanomaterials, 2021, doi:10.3390/nano11102560_

Round 1
Reviewer 1 Report
I carefully read this new revised version and thanks to the introduction of specific test for the functional properties of the new material, the paper is now suitable for publication in this form without further revision.
Reviewer 2 Report
1. The SEM investigation of the hemp fabric sample should be provided. 2. To make it more concise, the Table 1, 2 and 3 can be combined in one table. 3. Line 252, “the formation of Ag3+/citrate complexes”, the Ag ion cannot has 3+. 4. According to the Figure 10, the UV-Vis absorbance signal of sample F is too weak to make convincing results. 5. Some peaks attribution of the FTIR spectrum of the dry yeast sample are incorrect, such as 3417 cm-1, 1631 cm-1. 6. Many references have mall errors in format such as Ref. 1, 17, 18 et al.Author Response
Please see the attachment

This manuscript is a resubmission of an earlier submission. The following is a list of the peer review reports and author responses from that submission.
Round 1
Reviewer 1 Report
Overall, the mauscript has been improved. However, there are some points, which should be clarity. However, there are some points, which should be clarity.
- The English should be carefully revised throughout a manuscript.
- Conclusion should be carefully rewritten
This manuscript can be considered for publication only when the above-mention questions were especially stressed in the revised manuscript. The referee would like to review a revised version of this paper in the future.
Reviewer 2 Report
I carefully read the revised manuscript, but, even if the authors added the UV-Vis measurements as supporting characterization of their Silver-citrate composites, nevertheless in my opinion the scientific sound of the paper is still very poor, limited only to the synthetic procedure and without possible applications for this material. As I have previously noted, this lack is the focus of the paper. I strongly suggest again to the author to find a real and suitable application for this material and rewrite the paper.
I confirm that in this version the paper is not suitable for publication in this journal
Reviewer 3 Report
Dear Authors,
in my opinion, your manuscript entitled: "Study of physico-chemical interactions during the production of silver citrate nanocomposites with hemp fiber" can be published in Nanomaterials journal after major revision.
Please, you can find the main remarks and comments below.
Introduction
You have to add more information about the techniques which are applied to the control of the oxidation state of noble metals, especially silver.
Materials and methods
You have to add information about the purity of reagents which were used in these studies.
If you describe that you used EDX analysis to estimate the chemical composition of materials you have to show the STEM images with marked areas which were analyzed using the EDX technique and the EDX spectra.
You have to add the information about the preparation of samples for UV-vis and STEM-EDXS measurements and the conditions of these measurements, e.g. the information about the reference materials which were used for UV-vis studies.
You have to connect the points: 2. Materials and methods and 3. Experimental procedures and the part of results can be published in the Supplementary Information.
Results and discussion
In the beginning, you write: "After these preparatory steps, target EDX analysis shows a composition in atomic percentage of 81.84 % Silver, 17.40% Nickel, and 0.76% iron on some area; 88.26 % Silver, 10.32 % Nickel,
0.44% iron on other area and even 100% Silver on some areas" (page no. 3), and then: "Analyzing the ablated area, an important increase in iron and nickel is noticed. The elemental composition for two analyzed areas resulted to be 64.56% Silver, 27.01% Nickel, 8.43% Iron and 77.20 % Silver, 17.56 % Nickel, 5.20 % Iron, respectively". At the final, you show the data in Tables 1-3 for 3 different samples. It is very difficult to compare presented results because the data in the manuscript text do not describe the data in the tables. You have to show for each result obtained using EDX analysis: STEM image with the marked area which was analyzed, EDX spectra, and the information about the name of the sample. The oxidation state of silver species on the external surface of materials can be estimated using X-ray photoelectron spectroscopy (XPS). This technique can be used to estimate the chemical composition of materials.
You have to record and show the UV-vis spectra recorded for samples: citric acid (without silver))/glass.
If you want to know the oxidation state of silver species, you can use temperature-programmed reduction by hydrogen (TPR by H2), temperature-programmed deposition (TPD) of probe molecules, eg. NH3, CO, and/or the FTIR spectroscopy combined with the adsorption and desorption of probe molecules, e.g. CO.
Other remarks
The manuscript requires moderate changes in the English language.
You have to edit the description of references according to the template for Nanomaterials journal.
Kind regards,
Reviewer
Round 2
Reviewer 2 Report
I have evaluated the manuscript, and even if I have appreciated the revisions made, my opinion is still that the manuscript could be properly improved adding some issues about the functional properties of the fabricated materials. For example, antibacterial behaviuor could be evaluated, compared with some similar Ag-based composites.This adding, in addition to the fabrication method (by PLD) and characterizations that confirm the success of the procedure, could make the paper much more interesting for the scientific community.
So, taking in consideration these issues, in my opinion, after major revisions concerned the adding of at least anitbacterial properties, the paper could be suitable for publication.
Reviewer 3 Report
Dear Authors,
in my opinion, your corrected manuscript entitled: "Study of physico-chemical interactions during the production of silver citrate nanocomposites with hemp fiber" can be published in Nanomaterials after minor revision.
You have to edit your corrected manuscript according to the template of Nanomaterials journal.
Kind regards,
Reviewer
Round 3
Reviewer 2 Report
I understand the reasons about the lacking of time for carrying out properly the suggested test, but, even the perspective to future measurements is quite reassuring, nevetheless, in my opinion some demonstrations about the effectiveness in functional properties for a new synthesized material is mandatory.
For this reason I again strongly suggest to the authors to re-submit the paper after having these results